# A Zeroth-Order Adaptive Learning Rate Method to Reduce Cost of Hyperparameter Tuning for Deep Learning

Yanan Li [1,2], Xuebin Ren [2,3], Fangyuan Zhao [2,3] and Shusen Yang [1,2,*]

1    School of Mathematics and Statistics, Xi'an Jiaotong University, Xi'an 710049, China; gogll2@stu.xjtu.edu.cn
2    National Engineering Laboratory for Big Data Analytics, Xi'an Jiaotong University, Xi'an 710049, China; xuebinren@mail.xjtu.edu.cn (X.R.); zfy1454236335@stu.xjtu.edu.cn (F.Z.)
3    School of Computer Science and Technology, Xi'an Jiaotong University, Xi'an 710049, China
*    Correspondence: shusenyang@mail.xjtu.edu.cn

**Abstract:** Due to powerful data representation ability, deep learning has dramatically improved the state-of-the-art in many practical applications. However, the utility highly depends on fine-tuning of hyper-parameters, including learning rate, batch size, and network initialization. Although many first-order adaptive methods (e.g., Adam, Adagrad) have been proposed to adjust learning rate based on gradients, they are susceptible to the initial learning rate and network architecture. Therefore, the main challenge of using deep learning in practice is how to reduce the cost of tuning hyper-parameters. To address this, we propose a heuristic zeroth-order learning rate method, *Adacomp*, which adaptively adjusts the learning rate based only on values of the loss function. The main idea is that Adacomp penalizes large learning rates to ensure the convergence and compensates small learning rates to accelerate the training process. Therefore, Adacomp is robust to the initial learning rate. Extensive experiments, including comparison to six typically adaptive methods (Momentum, Adagrad, RMSprop, Adadelta, Adam, and Adamax) on several benchmark datasets for image classification tasks (MNIST, KMNIST, Fashion-MNIST, CIFAR-10, and CIFAR-100), were conducted. Experimental results show that Adacomp is not only robust to the initial learning rate but also to the network architecture, network initialization, and batch size.

**Keywords:** deep learning; adaptive learning rate; robustness; stochastic gradient descent

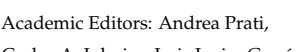



## 1. Introduction

Deep learning has been highly successful across a variety of applications, including speech recognition, visual object recognition, and object detection [1–3]. In general application, deep learning consists of training and inference phases. In the training phase, a predefined network is trained on a given dataset (known as the training set) to learn the underlying distribution characteristics. In the inference phase, the well-trained network is then used for unforeseen data (known as the test set) to implement specific tasks, such as regression and classification. One fundamental purpose of deep learning is to achieve as high accuracy as possible in the inference phase after only learning from the training set. In essence, training a deep learning network is equivalent to minimizing an unconstrained non-convex but smooth function.

$$\min_{w \in \mathbb{R}^n} f(w) := \mathbb{E}_{\xi \sim \mathcal{D}} F(w, \xi) \tag{1}$$

where $\mathcal{D}$ is the population distribution, $F(w, \xi)$ is the loss function of sample $\xi$, and $f(w)$ is the expectation of $F(w, \xi)$ with respect to $\xi$. One effective method to solve problem (1)

is mini-batch stochastic gradient descent (SGD) [4]. That is, in each iteration $t$, the model parameter $w_t$ is updated to $w_{t+1}$, following

$$w_{t+1} = w_t - \gamma_t \frac{1}{|\mathcal{B}_t|} \sum_{\xi \in \mathcal{B}_t} \nabla F(w_t, \xi_i), \tag{2}$$

where $\gamma_t$ is the learning rate and $\mathcal{B}_t$ is a random mini-batch of size $b$. Here, based on three advantages discussed below, we consider using problem Equations (1) and (2) to analyze deep learning. *First*, generality. Because most deep learning networks correspond to non-convex optimization, the derived results for problem (1) can be applied to general deep learning tasks in practice. *Second*, effectiveness. Because the data scale used in deep learning is usually huge, Equation (2) can achieve a better utility-efficiency tradeoff than SGD or batch GD. *Third*, simplicity. When using Equation (2), searching for a solution to problem (1) is reduced to setting a proper learning rate $\gamma_t$.

Therefore, the research question is how to set a proper $\gamma_t$ in Equation (2) to ensure the convergence of problem (1). Without of loss generality, we make the following assumptions.

- Non-convex but smooth $f(w)$. That is, $f(w)$ is non-convex but satisfies $f(w_{t+1}) - f(w_t) \leq \langle \nabla f(w_t), w_{t+1} - w_t \rangle + \frac{L}{2} \|w_{t+1} - w_t\|^2$.
- Unbiased estimate and bounded variance of $g(w) \triangleq \nabla F(w, \xi)$. That is, $\mathbb{E}_{\xi \sim \mathcal{D}}[g(w)] = \nabla f(w)$, and $\mathbb{E}_{\xi \sim \mathcal{D}}[\|g(w) - \nabla f(w)\|^2] \leq \sigma^2$.

It is well-known that searching for the global minima of Equation (1) is NP-hard [5]. Therefore, one usually aims to search for the first-order stationary point, the gradient of which satisfies $\|\nabla f(w)\| < \varepsilon$ where $\varepsilon$ is a given error bound. For simplicity, denote the average of gradients on mini-batch $\mathcal{B}_t$ as $\bar{g}_t = \frac{1}{|\mathcal{B}_t|} \sum_{\xi \in \mathcal{B}_t} \nabla F(w_t, \xi_i)$. It is well-known that SGD can converge for proper settings of $\gamma_t$ [6–9]. Actually, based on the above assumptions and through direct calculation, we have

$$\sum_{t=1,\cdots,T} \left( \gamma_t - \frac{L\gamma_t^2}{2} \right) \mathbb{E}\|\nabla f(w_t)\|^2 \leq f(w_1) - f^* + \frac{L\sigma^2 \sum_{t=1}^T \gamma_t^2}{2b},$$

where $f^* = \min_{w \in \mathbb{R}^n} f(w)$ and $L, \sigma^2$ are defined in the assumptions. Therefore, for any given constant $\gamma = \min\{1/L, O(1/\sqrt{L+T})\}$, we can deduce that $\sum_{t=1}^T \mathbb{E}\|\nabla f(w_t)\|$ is bounded, which implicates that $\|\nabla f(w_t)\| \to 0$ as $t \to \infty$. That is, SGD is convergent and one can output the stationary solution with high probability [6].

Nevertheless, one challenge of applying these theoretical results in practice comprises the unknown parameters $f(w_1)$, $L$ and $\sigma$, which are related to network architecture, network initialization, loss function, and data distribution. To avoid computing exact values that are network- and dataset-dependent, there are two common ways to set the learning rate in practice. One way is setting an initial level at the beginning and then adjusting it with a certain schedule [4], such as step, multi-step, exponential decaying, or cosine annealing [10]. However, setting the learning rate typically involves a tuning procedure in which the highest possible learning rate is chosen by hand [11,12]. Besides, there are additional parameters in the schedule that also need to be tuned. To avoid the delicate and skillful parameter tuning, the other way is using adaptive learning rate methods, such as Adagrad [13], RMSprop [14], and Adam [15], in which only the initial learning rate needs to be predefined. However, as shown in our experiments and other studies [16–18], they are sensitive to the initial learning rate and each of them has its own effective intervals (refer to Figure 1). Usually, the setting of an initial learning rate is model- and dataset-dependent. This increases the cost of tuning the learning rate and the difficulty of selecting the proper adaptive method in practice.

This motivates us to design an adaptive method, which can reduce the cost of tuning the initial learning rate. Furthermore, the method should achieve a satisfied accuracy no matter what the network architecture and data are. To achieve this, we propose a zeroth-

order method, *Adacomp*, to adaptively tune the learning rate. Unlike existing first-order adaptive methods, which adjust learning rate by exploiting gradients or additional model parameters, Adacomp only uses the values of loss function and is derived from minimizing Equation (1) with respect to $\gamma_t$, which has the original expression

$$\gamma \leftarrow \gamma/2 + \frac{\max\{0, f(w_t) - f(w_{t+1})\}}{\theta\|g(w_t)\|^2}, \tag{3}$$

where $\theta$ is an undetermined parameter and $\theta\|g(w_t)\|^2$ will be further substituted by other explicit factors. Refer to Equation (8) for details. Note that Equation (3) only uses the observable variables $\|g(w_t)\|$ and $f(w_t) - f(w_{t+1})$ to adjust $\gamma$. It can be interpreted that when $f(w_k) - f(w_{k+1})$ dominates $(\gamma/2)\|g(w_t)\|^2$, we use an aggressive learning rate to enhance the progress. In contrast, we use an exponential decaying learning rate to ensure convergence. Therefore, in a highly abstract level, $\gamma$ is complementary to loss difference $f(t_k) - f(w_{t+1})$ and we name it *Adacomp*, which has the following two advantages. Firstly, Adacomp is insensitive to learning rate, batch size, network architecture, and network initialization. Secondly, due to only exploiting values of loss function rather than high-dimensional gradients, Adacomp has high computation efficiency.

In summary, our contributions are as follows.

- We propose a highly computation-efficient adaptive learning rate method, Adacomp, which only uses loss values rather than exploiting gradients as other adaptive methods. Additionally, Adacomp is robust to initial learning rate and other hyper-parameters, such as batch size and network architecture.
- Based on the analysis of Adacomp, we give a new insight into why a diminishing learning rate is necessary when solving Equation (1) and why a gradient clipping strategy can outperform a fixed learning rate.
- We conduct extensive experiments to compare the proposed Adacomp with several first-order adaptive methods on MNIST, KMNIST, Fashion-MNIST, CIFAR-10, and CIFAR-100 classification tasks. Additionally, we compare Adacomp with two evolutionary algorithms on MNIST and CIFAR-10 datasets. Experimental results validate that Adacomp is not only robust to initial learning rate and batch size, but also network architecture and initialization, with high computational efficiency.

The remainder is organized as follows. Section 2 introduces related work about typical first-order adaptive methods. Section 3 presents the main idea and formulation of Adacomp. In Section 4, we conduct extensive experiments to validate Adacomp, in terms of robustness to learning rate, network architectures, and other hyperparameters. We conclude the paper and list future plans in Section 5.

## 2. Related Work

There are many modifications to the gradient descent method, and the most powerful is Newton's method [19]. However, the requirements of a Hessian matrix and its inverse are prohibitive to compute in practice for large-scale models. Therefore, many first-order iterative methods have been proposed to either exploit gradients or to approximate the inverse of the Hessian matrix.

### 2.1. Learning Rate Annealing

One simple extension of SGD is the mini-batch SGD, which can reduce the variance by increasing the batch size. However, the proper learning rate is hard to set beforehand. A learning rate that is too small will slow down the convergence, while if it is too large it will cause large oscillation or even divergence. The ordinary method is adjusting the learning rate in the training process, such as by using the simulated annealing algorithm [20,21], or decreasing the learning rate when the loss value is less than a given threshold [22,23]. However, the iteration number and threshold must be predefined. Therefore, the method is not adjustable to different datasets.

### 2.2. Per-Dimension First-Order Adaptive Methods

In mini-batch SGD, a single global learning rate is set for all dimensions of the parameters, which may not be optimal when training data are sparse and different coordinates vary significantly. A per-dimension learning rate that can compensate for these differences is often advantageous.

Momentum [24,25] is one method of speeding up training per dimension. The main idea is to accelerate progress along dimensions in which gradients are in the same direction and to slow progress elsewhere. This is done by keeping track of past parameters with an exponential decay, $v_t = \rho v_{t-1} + \gamma \bar{g}_t, w_{t+1} = w_t - v_t$, where $\rho$ is an undetermined parameter to control the decay of previous updated parameters. This gives an intuitive improvement over SGD when the cost surface is a long narrow valley.

Adagrad [13,26] adjusts the learning rate according to the accumulated gradients and has shown significant improvement on large-scale tasks in a distributed environment [27]. This method only uses information of gradients with the following update rule: $w_{t+1} = w_t - \frac{\gamma}{\sqrt{\sum_{\tau=1}^{t} \bar{g}_\tau^2}} \cdot \bar{g}_t$.

Here the denominator computes the $L_2$ norm of all previous gradients per-dimension. Since the learning rate for each dimension inversely grows with the gradient magnitudes, the large gradients have a small learning rate and vice versa. This has the nice property, as in second-order methods, that the progress along each dimension evens out over time. However, Adagrad is sensitive to initial model parameters. When some dimensions of gradient are too large, or as gradients accumulate, the learning rate will quickly tend to zero before achieving a good result.

Subsequently, several adaptive methods are proposed to overcome the drawback.

RMSprop [14,28] is a modification of Adagrad, using the root-mean-square (RMS) to replace the denominator in Adagrad, with the following update rule:

$$\mathbb{E}[\bar{g}^2]_t = \rho \mathbb{E}[\bar{g}^2]_{t-1} + (1-\rho)\bar{g}_t^2, w_{t+1} = w_t - \frac{\gamma}{\sqrt{\mathbb{E}[\bar{g}^2]_t} + \varepsilon} \bar{g}_t.$$

This can mitigate the fast decay of learning rate in Adagrad and damp oscillation. The parameter $\rho$ is used to control the decaying speed and $\varepsilon$ is a positive small constant to keep the denominator meaningful.

Adadelta [29,30] is another improvement of Adagrad from two aspects. On one hand, Adadelta replaces the denominator in Adagrad by the average of exponential decay of history gradients over a window with some fixed size. On the other hand, Adadelta uses the Hessian approximation to correct units of updates. Adadelta uses the following update rule:

$$\mathbb{E}[\bar{g}^2]_t = \rho \mathbb{E}[\bar{g}^2]_{t-1} + (1-\rho)\bar{g}_t^2, \ \mathbb{E}[\Delta w^2]_t = \rho \mathbb{E}[\Delta w^2]_{t-1} + (1-\rho)\Delta w_t^2,$$

$$w_{t+1} = w_t - \frac{\sqrt{\mathbb{E}[\Delta w^2]_t}}{\sqrt{\mathbb{E}[\bar{g}^2]_t} + \varepsilon} \cdot \bar{g}_t.$$

The advantage is no requirement of manually setting the global learning rate. At the beginning and middle phases, Adadelta achieves a good accelerating effect. However, at the late phase, it may oscillate at local minima.

Adam [15,31] can be viewed as a combination of Momentum and RMSprop, with additional bias correction. Adam uses the following update rule:

$$m_t = \rho_1 m_{t-1} + (1-\rho_1)\bar{g}_t, v_t = \rho_2 v_{t-1} + (1-\rho_2)\bar{g}_t^2,$$

$$m_t \leftarrow \frac{m_t}{1-\rho_1^t}, v_t \leftarrow \frac{v_t}{1-\rho_2^t}, w_{t+1} = w_t - \gamma \frac{m_t}{\sqrt{v_t} + \varepsilon}.$$

Here $m_t, v_t$ are factors of momentum and root-mean-square, and $\frac{m_t}{1-\rho_1^t}, \frac{v_t}{1-\rho_2^t}$ are corresponding bias corrections.

Adamax [15] is an extension of Adam by generalizing the $L_2$ norm to the $L_p$ norm and letting $p \to +\infty$, with the update rule:

$$m_t = \beta_1 m_{t-1} + (1 - \beta_1)\bar{g}_t, v_t = \max\{\beta_2 \cdot v_{t-1}, \|\bar{g}_t\|_\infty\}, w_{t+1} = w_t - \frac{\gamma_t}{(1 - \beta_1^t)} \cdot \frac{m_t}{v_t}$$

Note that the difference between Adamax and Adam is the expression of $v_t$ and there is no bias correction in $v_t$.

However, all these adaptive methods rely on gradients or additional model parameters, which make them sensitive to the initial learning rate and network architectures.

### 2.3. Hyperparameter Optimization

This aims to find the optimal learning rate values and includes experimental performance analysis [18,32] and Bayesian optimization [33,34] based on mathematical theory. Specifically, [35] combines hyperband and Bayesian optimizations, additionally utilizing the history information of previous explored hyperparameter configurations to improve model utility. Besides, reinforcement learning (RL) and heuristic algorithms are also extensively applied to tune hyperparameters of deep learning. With respect to RL, the focus is how to alleviate the dependence on expert knowledge [16,36,37] or additionally improve the computational efficiency [38,39]. For example, [16] uses RL to learn and adjust the learning rate on each training step and [36] proposes an asynchronous RL algorithm to search the optimal convolutional neural network (CNN) architecture. Ref. [39] reuses the previously successful configuration for reshaping the advantage function and [38] adaptively adjusts the horizon of the model to improve the computational efficiency of RL. With respect to heuristic algorithms, Ref. [40] sets an adaptive learning rate for each layer of neural networks by simulating the cross-media propagation mechanism of light in the natural environment. Ref. [41] proposes to use a variable length genetic algorithm to tune the hyperparameters of a CNN. Ref. [42] proposes a multi-level particle swarm optimization algorithm for a CNN, where an initial swarm at level-1 optimizes architecture and multiple swarms at level-2 optimize hyperparameters. Ref. [43] combines six optimization methods to improve the performance of forecasting short-term wind speed, where local search techniques are used to optimize the hyperparameters of a bidirectional long short-term memory network. However, these methods suffer unavoidably from more time-consuming space searching.

### 2.4. Zeroth-Order Adaptive Methods

Like our method, Refs. [23,44] propose two methods that adjust the learning rate based on the training loss. In [23], the learning rate at epoch $s$ is set as $\gamma_s = \gamma_0 \Pi_{t=1}^{s-1} r_{n_t}$, where $r_{n_t}$ is the scale factor. However, the selection of $r_{n_t}$ is through multi-point searching (controlled by beam size), which may be a computation overhead. For example, when the beam size is 4, it has to train the network with 12 different learning rates and to select the optimal one after the current epoch. In [44], the learning rate update is $\gamma_{t+1} = (1 + \mu)\gamma_t$ with $\mu = M \cdot (1 - e^{E(t+1)-E(t)})$, where $M$ is the tracking coefficient and $E(\cdot)$ is the reconstruction error of the WAE (wavelet auto-encoder). Note that the update rule is model restricted and cannot be applied to general cases. For example, when $E(t+1) - E(t) > 0$, $\gamma_{t+1}$ may be less than zero and has no meaning. However, the computation overhead of [23] and no meaning of [44] will not occur in Adacomp.

## 3. Adacomp Method

In the section, we describe our proposed Adacomp through two main steps. First, we deduce the optimal learning rate in each iteration based on theoretical analysis. Then, we design the expression of Adacomp to satisfy several restrictions between learning rate and difference of loss function. For convenience, we summarize the above mentioned and the following used variables in Table 1.

**Table 1.** List of variables and explanations.

| Variables | Explanations |
|---|---|
| $T, t$ | Number of total iterations and index of the current iteration. |
| $F(w, \xi), \nabla F(w, \xi)$ | Empirical loss function and gradient at model parameter $w$ and sample $\xi$. |
| $f(w), \nabla f(w)$ | Expected function and gradient of $F(w, \xi)$ and $\nabla F(w, \xi)$ with respect to $\xi \sim \mathcal{D}$. |
| $L, \sigma^2$ | Smooth constant of $f(w)$ and variance of $\nabla F(w, \xi)$ with respect to $\xi$. |
| $g_t, \bar{g}_t$ | Abbreviations for $\nabla F(w_t, \xi)$ and $\frac{1}{b} \sum_{\xi \in \mathcal{B}_t} \nabla F(w_t, \xi)$, respectively. |
| $\mathcal{B}_t, b$ | Mini-batch sampled at $t$-th iteration with size of $b$. |
| $\gamma_t, \beta$ | Learning rate at $t$-th iteration, parameter of Adacomp used to adjust $\gamma_t$. |
| $\Delta_t$ | Difference in $f(w)$ at two consecutive iterations, i.e., $f(w_{t-1}) - f(w_t)$. |

*3.1. Idea 1: Search Optimal Learning Rate*

For problem (1) in which gradients are Lipschitz continuous, i.e., $f(w_{t+1}) - f(w_t) \leq \langle \nabla f(w_t), w_{t+1} - w_t \rangle + \frac{L}{2} \|w_{t+1} - w_t\|^2$, by substituting Equation (2), we have

$$f(w_{t+1}) - f(w_t) \leq -\gamma_t \langle \nabla f(w_t), \bar{g}_t \rangle + \frac{L\gamma_t^2}{2} \|\bar{g}_t\|^2. \tag{4}$$

To make progress at each iteration $t$, let the r.h.s. of Equation (4) be less than zero, and we have $\gamma_t \leq 2\langle \nabla f(w_t), \bar{g}_t \rangle / L\|\bar{g}_t\|^2$. To greedy search the optimal learning rate, we minimize Equation (4) to obtain

$$\gamma_t = \langle \nabla f(w_t), \bar{g}_t \rangle / (L\|\bar{g}_t\|^2). \tag{5}$$

This presents an explicit relation between the learning rate and inner product of expected gradient $\nabla f(w_t)$ and observed gradients $\bar{g}_t$. When $\langle \nabla f(w_t), \bar{g}_t \rangle \geq 0$, Equation (5) is meaningful ($\gamma_t \geq 0$) and using this $\gamma_t$ can make the largest progress at the current iteration. Elsewhere, we set $\gamma_t = 0$ when $\langle \nabla f(w_t), \bar{g}_t \rangle \leq 0$. This means that when $\bar{g}_t$ is far away from the correct direction $\nabla f(w_t)$, we will drop the incorrect direction and do not update the model parameters at the current iteration.

By substituting Equation (5) into Equation (4) and through simple calculation, we obtain Theorem 1, which indicates that $\|\nabla f(w_t)\| \to 0$ as $t \to \infty$, i.e., the convergence of nonconvex problem (1).

**Theorem 1.** *If the learning rate $\gamma_t$ is set as Equation (5), then*

$$\sum_{t=1}^{T} \|\nabla f(w_t)\|^2 \leq 2L\sigma^2(f(w_1) - f^*)/b \tag{6}$$

*where $f^* = \inf_{w \in \mathbb{R}^n} f(w)$.*

**Proof.** By substituting Equation (5) into Equation (4), we have

$$f(w_{t+1}) - f(w_t) \leq -\frac{\langle \nabla f(w_t), \bar{g}_t \rangle^2}{2L\|\bar{g}_t\|^2}.$$

Taking expectation with respect to $\xi$ on the current condition $w_t$, we have

$$\mathbb{E}[f(w_{t+1}) - f(w_t)] \overset{(a)}{\leq} -\frac{\langle \nabla f(w_t), \mathbb{E}\bar{g}_t \rangle^2}{2L\mathbb{E}\|\bar{g}_t\|^2} \overset{(b)}{\leq} -\frac{\|\nabla f(w_t)\|^4}{2L(\|\nabla f(w_t)\|^2 + \sigma^2/b)} \overset{(c)}{\leq} -\frac{\|\nabla f(w_t)\|^2}{2L\sigma^2/b},$$

where we use Jensen's inequality $\mathbb{E}\langle \nabla f(w_t), \bar{g}_t \rangle^2 \geq \langle \nabla f(w_t), \mathbb{E}\bar{g}_t \rangle^2$ in (a) and assumption of bounded variance $\mathbb{E}\|\bar{g}_t\|^2 \leq \|\nabla f(w_t)\|^2 + \frac{\sigma^2}{b}$ in (b), and the fact $\frac{\|\nabla f(w_t)\|^2}{\|\nabla f(w_t)\|^2 + \sigma^2} \geq \frac{1}{\sigma^2}$ in

(c). Taking summation on both sides of the above inequality from $t = 1$ to $T - 1$ and using the fact that $f(w_1) - f(w_T) \le f(w_1) - f*$ derive the proved result. □

Some remarks about Theorem 1 are in order.

First, Theorem 1 achieves the optimal convergence rate of smooth nonconvex optimization when using the SGD algorithm. In particular, under a deterministic setting (i.e., $\sigma = 0$), Nesterov [45] shows that after running the method for at most $T = O(1/\varepsilon)$ steps, one can achieve $\min_{t=1,\cdots,T} \|\nabla f(w_t)\| \le \varepsilon$, where $\varepsilon$ is the given error bound. Under a stochastic setting (i.e., $\sigma > 0$), the result reduces to $O(1/\varepsilon^2)$. Ghadimi [6] derives the result $O(1/\varepsilon + 1/\varepsilon^2)$ of randomly outputting a solution $w_R$ satisfying $\mathbb{E}\|\nabla f(w_R)\|^2 \le \varepsilon$ with high probability, where $R \in [T]$. The optimal rate is improved to $O(\varepsilon^{-7/4} \log(1/\varepsilon))$ by using accelerated mirror descent [7]. Here, we obtain that using Equation (6) achieves the optimal convergence rate, i.e., $O(\varepsilon^{-2})$, for stochastic optimization when using SGD.

Second, a diminishing learning rate is necessary to ensure convergence, i.e., gradient $\|\nabla f(w_t)\|$ will tend to zero as $t \to \infty$. In this case, we have $\langle \nabla f(w_t), \bar{g}_t \rangle \to 0$. Additionally, because $\mathbb{E}\|\bar{g}_t\|^2 = \|\nabla f(w_t)\|^2 + \sigma^2 \to \sigma^2$, we deduce that $\gamma_t = \langle \nabla f(w_t), \bar{g}_t \rangle / (L\|\bar{g}_t\|^2) \to 0$ holds almost surely.

Third, gradient clipping is a useful skill to ensure convergence. As proved in [46], using a clipped gradient can converge faster than a fixed learning rate. Here we show an explicit explanation. Based on Equation (5), $\gamma_t = \min\{ \frac{\langle \nabla f(w_t), \bar{g}_t \rangle}{L\|\bar{g}_t\|} / \|\bar{g}_t\|, \gamma_t \}$. Therefore, using Equation (5) is equivalent to using threshold $c = \frac{\langle \nabla f(w_t), \bar{g}_t \rangle}{L\gamma_t\|\bar{g}_t\|}$ to clip the gradient, which in turn means that clipping the gradient is equivalent to using a varying learning rate to update model parameters. Therefore, for any fixed learning rate $\gamma_t \equiv \gamma$, we can adjust it based on Equation (5) to make faster progress. That is, proper gradient clipping can outperform any fixed learning rate update.

### 3.2. Idea 2: Approximate Unknown Terms

To set learning rate according to Equation (5), two terms, expected gradient $\nabla f(w_t)$ and smooth constant $L$, are unknown in practice. Note that in the training process, we only get access to information of stochastic gradient $\bar{g}_t$, model parameters $w_t$, and loss value $f(w_t)$. Most first-order adaptive learning methods exploit stochastic gradient and model parameters, which are usually high-dimensional vectors. Instead, we will use the stochastic gradient $\bar{g}_t$ and loss value to reformulate Equation (5). In particular, based on Equation (4), we have

$$\gamma_t = \frac{\langle \nabla f(w_t), \gamma_t\bar{g}_t \rangle}{L\gamma_t\|\bar{g}_t\|^2} \le \frac{\frac{L\gamma_t^2}{2}\|\bar{g}_t\|^2 + f(w_t) - f(w_{t+1})}{L\gamma_t\|\bar{g}_t\|^2}. \tag{7}$$

In the above inequality, parameters square norm of gradients $\|\bar{g}_t\|^2$ and learning rate $\gamma_t$ are known, and loss value of expected $f(w_t)$ at the current iteration can be approximated by empirical value $F(w_t, \xi)$; however, smooth constant $L$ and the loss value $f(w_{t+1})$ of the next iteration are unknown. Derived from the basic analysis [6], it is known that when $L\gamma_t \le 1$, SGD can converge to the stationary point. Therefore, we introduce a new parameter $\theta := L\gamma_t \in (0, 1]$ and use $f(w_{t-1}) - f(w_t)$ to approximate $f(w_t) - f(w_{t+1})$ based on the assumption that $f(w)$ is smooth. Then, for any given learning rate $\gamma$ subject to $\theta \stackrel{\triangle}{=} L\gamma \in (0, 1]$, we adjust the learning rate based on the following formula:

$$\gamma \leftarrow \frac{\frac{\theta\gamma}{2}\|\bar{g}_t\|^2 + \max\{f(w_{t-1}) - f(w_t), 0\}}{\theta\|\bar{g}_t\|^2}, \theta \in (0, 1]. \tag{8}$$

Equation (8) has straightforward interpretations. *First*, when $\|\bar{g}_t\|^2$ dominates $f(w_{t-1}) - f(w_t)$ in the numerator, we will use an exponential decaying learning rate strategy to prevent divergence. This includes the case that $f(w_{t-1}) - f(w_t)$ tends to zero, i.e., the model converges to a stationary point. In such case, $\gamma_t$ will decay to zero to

stabilize the process. As shown in [46], in such case, any fixed large learning larger than a threshold will diverge the training. *Second*, when $f(w_{t-1}) - f(w_t)$ is relatively large, which means that the current model point is located at the rapid descent surface of $f(w)$, we can use a relatively large learning rate to accelerate the descent speed. This usually happens at the initial phase of the training procedure. *Third*, the parameter $\theta \in (0, 1]$ can control the tradeoff between $\|\bar{g}_t\|^2$ and $f(w_{t-1}) - f(w_t)$. Note that these two terms have different magnitudes, and $\|\bar{g}_t\|^2$ is generally much larger than $f(w_{t-1}) - f(w_t)$. In such case, Equation (8) with $\theta = 1$ reduces to a totally exponential decaying strategy, which may slow down the convergence. To address it, one can set an adaptive $\theta$ according to $f(w_{t-1}) - f(w_t)$.

Although Equation (8) is meaningful, the remaining challenge is how to set the proper $\theta$ according to the observed gradients $\|\bar{g}_t\|^2$ and difference in loss function $f(w_{t-1}) - f(w_t)$. To address this, we deal with $\theta\|\bar{g}_t\|^2$ as a whole and reformulate the term $\frac{\max\{f(w_{t-1}) - f(w_t), 0\}}{\theta\|\bar{g}_t\|^2}$ as a pure function of $f(w_{t-1}) - f(w_t)$. Here, we propose an effective adaptive schedule Adacomp that only uses the information of loss values and satisfies the following requirements.

- *When $f(w_{t-1}) - f(w_t) < 0$, we should decrease $\gamma$ to restrict the progress along the wrong direction.*
- *When $f(w_{t-1}) - f(w_t) > 0$, we should increase $\gamma$ to make aggressive progress.*
- *When $\gamma$ is too small, we should increase $\gamma$ to prevent Adacomp from becoming trapped in a local minimum.*
- *When $\gamma$ is too large, we should decrease $\gamma$ to stabilize the training process.*

These requirements motivate us to design Adacomp based on the *arctan* function, which is flexible to small values but robust and bounded to large values. The main principle is as follows:

*Decompose $\gamma$ into three parts, $\mathrm{II}_1$, $\mathrm{II}_{21}$, and $\mathrm{II}_{22}$. $\mathrm{II}_1$ is used to compensate the learning rate to accelerate training when the loss function decreases. However, the compensation should be inverse to learning rate and bounded. $\mathrm{II}_{21}$ and $\mathrm{II}_{22}$ are used to penalize the learning rate to stabilize training when the loss function increases, but with a bounded amplitude when the learning rate is too large or small.*

Based the above principle, we reformulate Equation (8) as:

$$\gamma \leftarrow \frac{\gamma}{2} + \gamma\mathbb{I}_{\Delta_t > \epsilon}\,\mathrm{II}_1 + \gamma\mathbb{I}_{\Delta_t < -\epsilon}(\mathrm{II}_{21} + \mathrm{II}_{22}), \tag{9}$$

where $\mathbb{I}$ is the indicator function of $\Delta_t \overset{\triangle}{=} f(w_{t-1}) - f(w_t)$ and $\epsilon$ is a small positive constant. Many functions satisfy the above discipline. To reduce the difficulty of designing $\mathrm{II}_1, \mathrm{II}_{21}, \mathrm{II}_{22}$, we define the following expressions, where only one parameter $\beta$ needs to be tuned. Furthermore, as experimental results show (refer to Figure A1), Adacomp is not so sensitive to $\beta$.

$$\mathrm{II}_1 = \frac{1}{2} + \frac{\arctan(\Delta_t + 1/\gamma)}{5\pi},\ \mathrm{II}_{21} = \frac{1}{4} - \frac{\arctan(\gamma)}{\beta\pi},\ \mathrm{II}_{22} = \frac{1}{4} - \frac{\arctan(-\Delta_t + 1/\gamma)}{5.5\pi},$$

where $1/2 < \beta \leq 5$ is a parameter used to control the adjustment amplitude of the learning rate.

We first explain the meaning of $\varepsilon$ and then $\mathrm{II}_1, \mathrm{II}_{21}$, and $\mathrm{II}_{22}$.

The meaning of $\epsilon$. We replace the hard threshold $\Delta_t < 0, > 0$ with soft threshold $\Delta_t < -\epsilon, > \epsilon$ on two aspect considerations. *First*, this can alleviate the impacts of $\Delta_t$'s randomness on learning rate. *Second*, when $\Delta_t \in [-\epsilon, \epsilon]$ for small positive values such as $\epsilon = 10^{-5}$, we halve $\gamma$. Actually, in such a case the training has converged and halving $\gamma$ can make training more stable.

Now, we explain how to set expressions of $\mathrm{II}_1, \mathrm{II}_{21}$, and $\mathrm{II}_{22}$.

- Expression of $\mathrm{II}_1$. Note that $\mathrm{II}_1$ works only if $\Delta_t > 0$. In this case, we should increase $\gamma$ to speed up training. However, the increment should be bounded and inverse

to current $\gamma$ to avoid divergence. Based on these considerations, we define $\mathrm{II}_1$ as $\mathrm{II}_1 = \frac{1}{2} + \frac{\arctan(\Delta_t + 1/\gamma)}{5\pi}$, where $1/2$ is used to keep $\gamma$ unchanged and $\arctan(\cdot)/(5\pi)$ is used to ensure that the increment amplitude is at most $\gamma/10$.

- Expressions of $\mathrm{II}_{21}$ and $\mathrm{II}_{22}$. Note that $\mathrm{II}_{21}, \mathrm{II}_{22}$ work only if $\Delta_t < 0$, where the twice $1/4$ is used to keep $\gamma$ unchanged and the remaining terms are used to control the decrement amplitude. In this case, we should decrease $\gamma$ to prevent too much movement along the incorrect direction. However, the decrement should be bounded to satisfy the two following properties.

  - When $\gamma$ is small, the decrement of $\mathrm{II}_{21} + \mathrm{II}_{22}$ should be less than the increment in $\mathrm{II}_1$ given the same $|\Delta_t|$ unless $\gamma$ is forced to zero, which potentially leads to training stopping too early. Therefore, we define $\frac{\arctan(\cdot)}{5.5\pi}$ in $\mathrm{II}_{21}$ to satisfy $\frac{\arctan(\cdot)}{5.5\pi} < \frac{\arctan(\cdot)}{5\pi}$. Note that $\mathrm{II}_{21} + \mathrm{II}_{22} \approx \frac{1}{2} - \frac{\arctan(-\delta_t + 1/\gamma)}{5.5\pi}$ when $\gamma$ is small, which satisfies the requirement.

  - When $\gamma$ is large, the decrement of $\mathrm{II}_{21} + \mathrm{II}_{22}$ should be larger than the increment in $\mathrm{II}_1$ given the same $|\Delta_t|$. That is, we can fast decrease the large learning rate when the function loss is negative. Therefore, we set $\beta \in (1/2, 5]$ in $\mathrm{II}_{21}$ to satisfy that $\frac{\arctan(\gamma)}{\beta\pi} + \frac{\arctan(-\Delta_t + 1/\gamma)}{5.5\pi}$ (in the case $\Delta_t < 0$) is greater than $\frac{\arctan(\Delta_t + 1/\gamma)}{5\pi}$ (in the case $\Delta_t > 0$).

Furthermore, the inferior bound $1/2$ of $\beta$ is used to ensure $\frac{\gamma}{2} + \gamma\mathbb{I}_{\Delta_t < -\epsilon}(\mathrm{II}_{21} + \mathrm{II}_{22}) > 0$ when $\Delta_t < -\epsilon$, i.e., the meaningfulness of $\gamma$.

Remark. In $\mathrm{II}_1, \mathrm{II}_{22}$, the denominator $5\pi, 5.5\pi$ can be generalized as $\alpha_1, \alpha_2$, with $\alpha_1 < \alpha_2$, while keeping the difference between $\mathrm{II}_1$ and $\mathrm{II}_{22}$ is not significant. Meanwhile, $1/2 < \beta \leq 5$ is replaced by $1/2 < \beta \leq \alpha_1$. Note that any settings satisfying the above requirements can be used to substitute Equation (9). Here, $5\pi$ and $5.5\pi$ are just one satisfied setting and one can adjust the parameter $\beta$ to control the tradeoff between $\mathrm{II}_1$ and $\mathrm{II}_{21} + \mathrm{II}_{22}$.

In summary, Equation (9) penalizes the large learning rate and compensates the too small learning rate, and presents an overall decreasing trend. These satisfy all the requirements. Note that this adjusting strategy is complementary to $\Delta_t$ and $\gamma$; we named it Adacomp. Although Adacomp is robust to hyperparameters, it may fluctuate at a local minimum. The reason for this is that Adacomp will increase the learning rate when the loss function decreases near the local minima, and the increment potentially makes model parameters skip the local minima. Then, Adacomp in turn will penalize learning rate to decrease loss function. Thus, the alternately adaptive adjustment possibly makes model parameters fluctuate at a local minimum. To achieve both robustness and high-utility, one can view Adacomp as a pre-processing step to reduce the cost of tuning hyperparameters, and combine Adacomp with other methods to improve the final model utility. Algorithm 1 shows the pseudocode, which consists of two phases. Phase one uses Adacomp to fast-train the model by avoiding and carefully tuning the initial learning rate. Phase two uses other methods (e.g., SGD) to improve the model utility. More details are as follows. At the beginning (Input), one should define four parameters, including number of total iterations $T$ with phase threshold $T_1$, initial learning rate $\gamma_0$, and one specific value $\beta$ of Adacomp used to control the tradeoff between increment and decrement.

In phase 1 (Lines 2–8), two variables , $loss_l, loss_c$, are used to track the loss difference $\Delta_t$ in Equation (9), where $loss_l = \frac{1}{b}\sum_{\xi \in \mathcal{B}_{t-1}} F(w_{t-1}, \xi)$ is loss of last time and $loss_c = \frac{1}{b}\sum_{\xi \in \mathcal{B}_t} F(w_t, \xi)$ is loss of current time (Lines 3–5). Then $\Delta_t$ is used to update learning rate based on Equation (9) (Line 6). The updated learning rate $\gamma_t$ is used to update the model parameters $w_t$ to $w_{t+1}$ based on $w_{t+1} = w_t - \gamma_t \bar{g}_t$ (Line 7) until $t$ reaches to $T_1$. In phase 2, we set the average of the learning rate used in phase one, $\gamma_{avg}$, as the new current initial learning rate (Line 9). Then, one can adopt a decaying scheduler, such as StepLR in mini-batch gradient descent update to stabilize the training process (Lines 11–13). The final model parameter is output when the stopping criterion is satisfied (Line 15).

---

**Algorithm 1** Two-phase Training Algorithm.

---

**Input:** Initial $\gamma_0$, balance rate $\beta$, iteration number $T$, and phase threshold $T_1$

**Output:** $\mathbf{w}_T$

1 Initialize loss variables $loss_l = 0$, $loss_c = 0$

  // Phase one: Adacomp

2 **for** $t = 1 : T_1$ **do**

3     | Compute gradient $\bar{g}_t$ and loss value $loss_t$;

4     | Compute $\Delta_t$ via:

5     | $loss_l \leftarrow loss_c$, $loss_c \leftarrow loss_t$, $\Delta_t = loss_l - loss_c$;

6     | Use $\Delta_t$ to update $\gamma_t$ based on Equation (9);

7     | Update $w_{t+1} = w_t - \gamma_t \bar{g}_t$;

8 **end**

  // Phase two: SGD with StepLR scheduler

9 Compute average of $\{\gamma_t\}$ and denote as $\gamma_{avg}$;

10 **for** $t = T_1 : T$ **do**

11     | Compute gradient $\bar{g}_t$;

12     | Update $w_{t+1} = w_t - \gamma_{avg} \bar{g}_t$;

13     | Decay $\gamma_{avg}$ with a given scheduler;

14 **end**

15 **return** $w_T$.

---

## 4. Experiments

In this section, we describe extensive experiments to validate the performance of Adacomp on five classification datasets, in comparison with other first-order iterative methods, including SGD, Momentum, Adagrad, RMSprop, Adadelta, Adam, and Adamax. Experimental results show that Adacomp is more robust to hyper-parameters and network architectures. Code and experiments are publicly available at https://github.com/IoTDATALab/Adacomp, (accessed on 26 October 2021).

### 4.1. Experimental Setup

We evaluated the eight first-order methods on five benchmark classification tasks, MNIST [47], KMNIST [48], Fashion-MNIST [49], CIFAR-10, and CIFAR-100 [50], which are used for 10 classification tasks, except CIFAR-100, which is used for 100 classification tasks. Moreover, MNIST, KMNIST, and Fashion-MNIST include 60,000 training samples and 10,000 test samples, while CIFAR-10 and CIFAR-100 include 50,000 training samples and 10,000 test samples. For MNIST, the network architecture used was borrowed from examples in Pytorch (https://github.com/pytorch/examples, (accessed on 15 May 2021)). In particular, it consists of two convolutionary layers (32, 64 channels with kernel size 3 and relu activation), followed by one max pooling layer with size 2 and two fully connected layers (the first layer with relu activation), ending with the *log_softmax* function. For CIFAR-10, we used 6 out of 18 network architectures (https://github.com/kuangliu/pytorch-cifar, (accessed on 15 May 2021)): LeNet, VGG19, ResNet18, MobileNet, SENet18, and SimpleDLA. For CIFAR-100, we used ResNet [51] and the code is available at https://github.com/junyuseu/pytorch-cifar-models.git, (accessed on 15 May 2021). All experiments were deployed on a local workstation with a 10-core Geforce GPU and 128 GB memory, and cuda version 11.0. Note that Adacomp can be implemented by directly replacing Equation (8) in Pytorch's SGD optimizer.

We set epochs as 10, 20, 100, and 150 for MNIST, KMNIST and Fashion-MNIST, CIFAR-10, and CIFAR-100, respectively. We set $T_1/T = 0.6$ in Algorithm 1.

### 4.2. Results on MNIST Dataset

We conducted Adacomp (set $\beta = 0.6$ in Equation (9)) on MNIST dataset to validate its robustness from aspects of learning rate, batch size, and initial model parameters.

Predominately, Adacomp works well for a wider range of learning rates than all other compared methods.

### 4.2.1. Robustness to Initial Learning Rate

We set learning rate (LR) from $10^{-5}$ to 15, and used fine granularity in the interval $[0.01, 1]$ but loose granularity outside it. Figure 1 shows the results of test accuracy after 10 epochs when LR lies in $[0.01, 1]$. It is observed that only Adadelta and Adacomp work when LR is greater than one. Additionally, we illustrate results when LR lies in $[1, 15]$ and the effective intervals of all methods are shown in Figure 2. The training batch size was fixed as 64 and the initial seed was set as 1 in all cases.

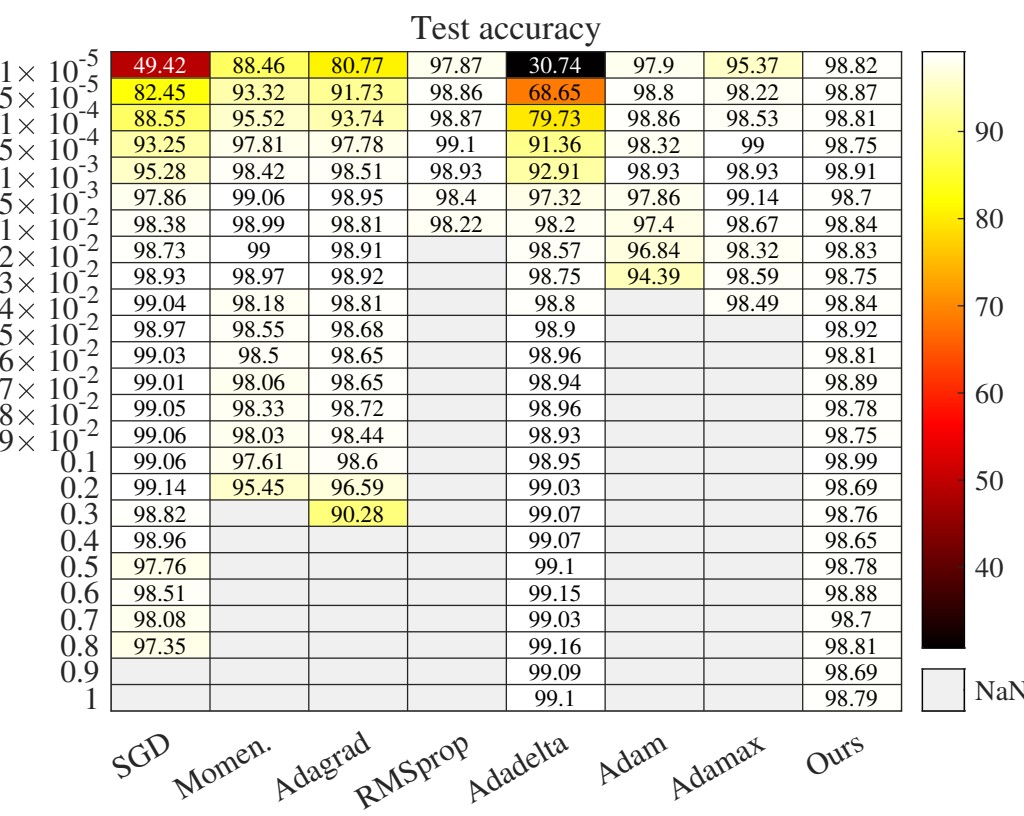

| LR | SGD | Momen. | Adagrad | RMSprop | Adadelta | Adam | Adamax | Ours |
|---|---|---|---|---|---|---|---|---|
| $1\times10^{-5}$ | 49.42 | 88.46 | 80.77 | 97.87 | 30.74 | 97.9 | 95.37 | 98.82 |
| $5\times10^{-5}$ | 82.45 | 93.32 | 91.73 | 98.86 | 68.65 | 98.8 | 98.22 | 98.87 |
| $1\times10^{-4}$ | 88.55 | 95.52 | 93.74 | 98.87 | 79.73 | 98.86 | 98.53 | 98.81 |
| $5\times10^{-4}$ | 93.25 | 97.81 | 97.78 | 99.1 | 91.36 | 98.32 | 99 | 98.75 |
| $1\times10^{-3}$ | 95.28 | 98.42 | 98.51 | 98.93 | 92.91 | 98.93 | 98.93 | 98.91 |
| $5\times10^{-3}$ | 97.86 | 99.06 | 98.95 | 98.4 | 97.32 | 97.86 | 99.14 | 98.7 |
| $1\times10^{-2}$ | 98.38 | 98.99 | 98.81 | 98.22 | 98.2 | 97.4 | 98.67 | 98.84 |
| $2\times10^{-2}$ | 98.73 | 99 | 98.91 | | 98.57 | 96.84 | 98.32 | 98.83 |
| $3\times10^{-2}$ | 98.93 | 98.97 | 98.92 | | 98.75 | 94.39 | 98.59 | 98.75 |
| $4\times10^{-2}$ | 99.04 | 98.18 | 98.81 | | 98.8 | | 98.49 | 98.84 |
| $5\times10^{-2}$ | 98.97 | 98.55 | 98.68 | | 98.9 | | | 98.92 |
| $6\times10^{-2}$ | 99.03 | 98.5 | 98.65 | | 98.96 | | | 98.81 |
| $7\times10^{-2}$ | 99.01 | 98.06 | 98.65 | | 98.94 | | | 98.89 |
| $8\times10^{-2}$ | 99.05 | 98.33 | 98.72 | | 98.96 | | | 98.78 |
| $9\times10^{-2}$ | 99.06 | 98.03 | 98.44 | | 98.93 | | | 98.75 |
| 0.1 | 99.06 | 97.61 | 98.6 | | 98.95 | | | 98.99 |
| 0.2 | 99.14 | 95.45 | 96.59 | | 99.03 | | | 98.69 |
| 0.3 | 98.82 | | 90.28 | | 99.07 | | | 98.76 |
| 0.4 | 98.96 | | | | 99.07 | | | 98.65 |
| 0.5 | 97.76 | | | | 99.1 | | | 98.78 |
| 0.6 | 98.51 | | | | 99.15 | | | 98.88 |
| 0.7 | 98.08 | | | | 99.03 | | | 98.7 |
| 0.8 | 97.35 | | | | 99.16 | | | 98.81 |
| 0.9 | | | | | 99.09 | | | 98.69 |
| 1 | | | | | 99.1 | | | 98.79 |

**Figure 1.** Test accuracy with respect to learning rate after 10 epochs. NaN denotes test accuracy that is less than 15%. Dataset is MNIST.

Two conclusions are observed from Figures 1 and 2.

First, Adacomp is much more robust than the other seven iterative methods. From Figure 1, Adacomp achieves test accuracy greater than 98% *for all settings* of learning rates. In contrast, RMSprop, Adam, and Adamax are sensitive to learning rate and only work well for very small learning rates. The other remaining methods, except Adadelta, have an intermediate robustness between Adamax and Adacomp. However, when the learning rate is greater than one, only Adadelta and Adacomp still work. Furthermore, from the top of Figure 2, it is observed that Adadelta works well until learning rate increases up to 9 while Adacomp still works even when learning rate is up to 15. The bottom of Figure 2 (logarithmic scale of horizontal axis) illustrates that each method has its own effective interval. From the top RMSprop to the penultimate Adadelta, the effective interval gradually slides from left to right. However, Adacomp (ours) has a much wider interval, which means one can successfully train the model almost without tuning learning rate. The reason for strong robustness is that Adacomp adjusts learning rate to a high value in the first few epochs before shrinking to a small value, no matter what the initial setting.

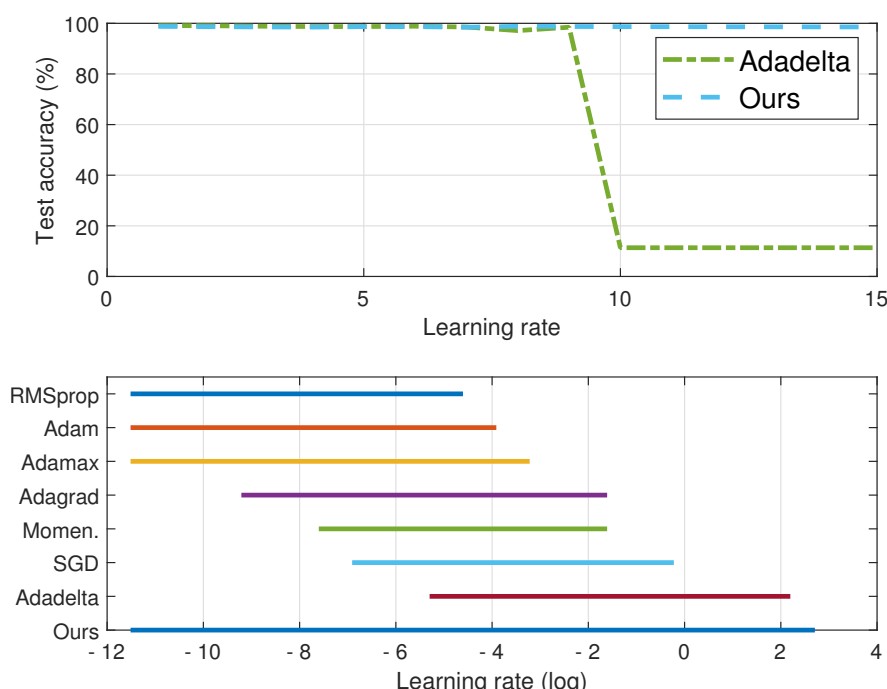

**Figure 2.** Comparison of test accuracy between Adadelta and Adacomp (**top** figure) and effective intervals (accuracy $\geq 95\%$) of eight iterative methods (**bottom** figure). Dataset is MNIST.

Second, the adaptive strategy may not always outperform SGD, which uses the fixed learning rate. As shown in Figure 1, adaptive methods, Momentum, Adagrad, RMSprop, Adam, and Adamax, are more sensitive to learning rate than SGD. This means that when using adaptive strategies, proper setting of the initial learning rate is necessary. However, we also observe that adaptive methods, Adadelta and Adacomp, are much more robust to initial learning rate than SGD.

### 4.2.2. Robustness to Other Hyperparameters

We conducted more experiments to compare the robustness of the eight iterative methods to batch sizes and initial model parameters. Figure 3a shows the impacts when batch sizes were set as 16, 32, 64, and 128, and Figure 3b shows the impacts when we repeated the experiment four individual times in which the network was initialized using the same random seed $(1, 10, 30, 50,$ respectively) for different optimizers. For fairness, we set learning rate to 0.01. Under the level, all eight methods have a roughly equivalent accuracy (refer to Figure 1). It is observed from Figure 3a that the robustness to batch size, from strong to weak, is in the order $Momentum \approx Adagrad \approx Adamax \approx Adacomp \succ Adadelta \succ SGD \succ RMSprop \succ Adam$ (note that Adam was divergent when the batch size was set to 16). It is observed from Figure 3b that the robustness to initialized seeds is in the order $Adagrad \approx Momentum \approx Adacomp \approx Adamax \succ Adadelta \succ SGD \succ Adam \succ RMSprop$ (note that RMSprop was divergent when the seed was set to $30, 50$). In summary, for a certain setting of learning rate, Momentum, Adagrad, Adamax, and Adacomp have the most robustness to batch sizes and initialized seeds, while RMSprop and Adam have the weakest robustness.

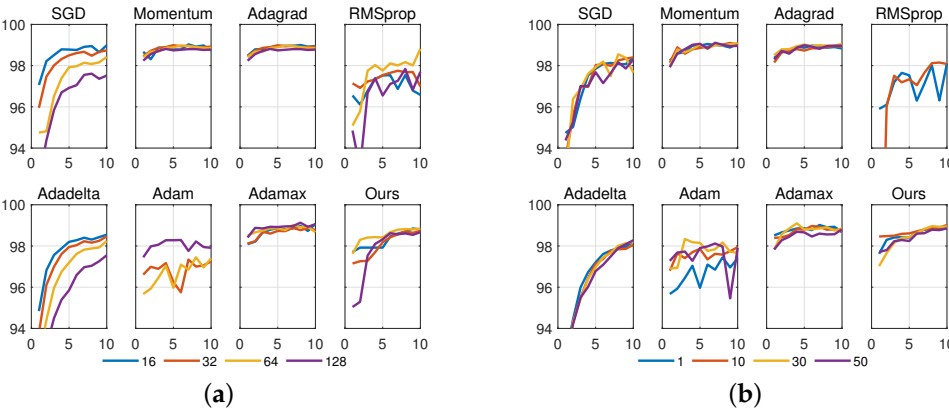

**Figure 3.** Robustness with respect to different batch sizes (16, 32, 64, 128 in left plot), and different initial seeds (1, 10, 30, 50 in right plot). Horizontal axis denotes the number of epochs. Dataset is MNIST. (**a**) Comparison with respect to batch size. Learning rate is fixed as 0.01 and initial seed is fixed as 1. (**b**) Comparison with respect to network initialization. Learning rate is fixed as 0.01 and batch size is fixed as 64.

### 4.2.3. Convergence Speed and Efficiency

To compare the convergence speed, we further recorded the number of epochs when prediction accuracy firstly grows up than 98% (Column 3 in Table 2). The convergence speed is defined as the ratio of Epochs to $\#_{\geq 98\%}$. Taking SGD as an example, it had 13 out of 25 times where the final prediction was greater than 98% (refer to Figure 1). Meanwhile, among the 13 successful cases, SGD took a total of 19 epochs to achieve 98%. Then, the convergence speed was $19/13 = 1.46$. It is observed from Table 2 that Adacomp has the fastest convergence speed 0.92, which means Adacomp can achieve at least an accuracy of 98% using no more than one epoch on average. Subsequently, Momentum has a convergence speed of 1.09 and Adamax has the slowest speed of 3.5.

To compare the efficiency, we performed each of the methods ten times and recorded the average training and total time in the last column of Table 2. It is also observed that Adacomp achieves the lowest time consumption (excluding SGD), while Momentum and RMSprop have a relatively large time consumption. Although the improvement is slight, it can be deduced that Adacomp performs fewer computations than other first-order adaptive methods to adjust learning rate.

**Table 2.** Number of epochs when each of the methods first grow greater than 98%. Speed is calculated as epochs divided by $\#_{\geq 98\%}$. Dataset is MNSIT.

| Method | $\#_{\geq 98\%}$ | Epochs | Speed | Training/Total Time (s) |
|---|---|---|---|---|
| SGD | 13 | 19 | 1.46 | 68.13/78.22 |
| Momentum | 11 | 12 | 1.09 | 69.11/79.36 |
| Adagrad | 12 | 22 | 1.83 | 68.55/78.68 |
| RMSprop | 6 | 16 | 2.67 | 69.18/79.34 |
| Adadelta | 19 | 43 | 2.26 | 68.56/78.68 |
| Adam | 4 | 7 | 1.75 | 68.40/78.49 |
| Adamax | 9 | 28 | 3.5 | 68.55/78.67 |
| Ours | 25 | 23 | 0.92 | 68.25/78.26 |

### 4.3. Results on CIFAR-10 Dataset

In this section, we conducted more experiments to further compare the robustness of eight iterative methods on CIFAR-10. Each of them was performed on three settings

of learning rate (i.e., 0.5, 0.05, and 0.005) and six network architectures (namely, LeNet, VGG, ResNet, MobileNet, SEnet, SimpleDLA). In this setting, we chose $\beta = 5$ in Adacomp. Figure 4 shows the comparison of test accuracy on six network architectures when LR is 0.5, 0.05, 0.005. Figure 5a,b show improvements in test accuracy when LR is degraded from 0.5 to 0.05, and 0.05 to 0.005, respectively. We conclude the robustness of eight methods with respect to different network architectures.

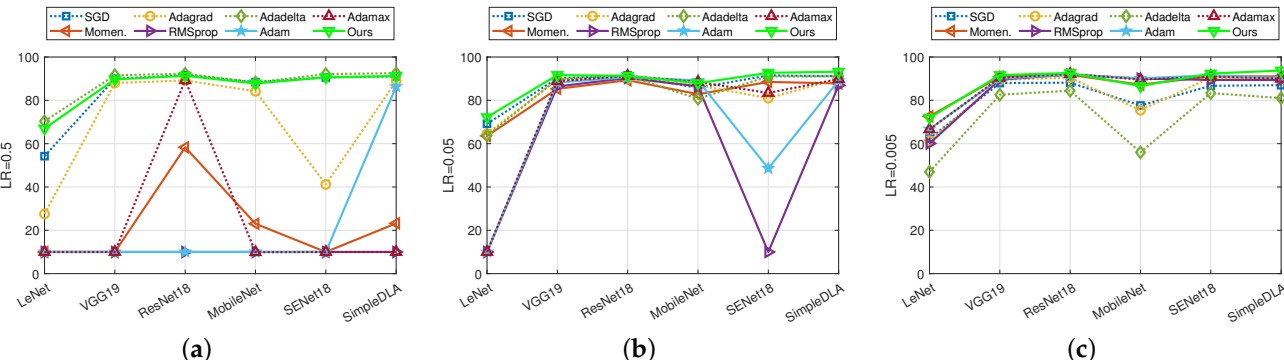

**Figure 4.** Accuracy comparison among six methods across six network architectures when learning rate was set to 0.5, 0.05, and 0.005. Dataset is CIFAR-10. (**a**) LR is 0.5; (**b**) LR is 0.05; (**c**) LR is 0.005.

Firstly, we separately conclude the model utility for each setting of learning rate. (1) From Figure 4a, it is observed that Adacomp has a similar performance to the best Adadelta. However, Momentum, RMSprop, Adam, and Adamax, are failures for almost all architectures when LR is 0.5. That is, these adaptive methods are not compatible with a large initial learning rate, which is also observed on MNIST (Figure 1). Among them, Adagrad is much more sensitive to the network architecture than Adadelta, Adacomp, and SGD. Surprisingly, SGD achieves a comparable result to Adadelta and Adacomp, except for the LeNet architecture. (2) From Figure 4b, it is observed that all methods, except RMSprop and Adam, have similar performances on each of the network architectures. Furthermore, except on the MobileNet architecture, Adacomp achieves the highest accuracy, followed by Adadelta and SGD (both have small degradation on SENet and SimpleDLA). Besides, it is observed that RMSprop and Adam are more sensitive to the network architectures than others. (3) From Figure 4c, it is observed that no method diverged when LR was set to 0.005. Specifically, adaptive methods, Momentum, RMSprop, Adam, Adamax, and Adacomp (ours) perform similarly on each of the six architectures, except that Adacomp has the advantage for SimpleLDA and disadvantage for MobileNet. Subsequently, Adagrad achieves a similar performance except on the MobileNet architecture. However, Adadelta and SGD have a total degradation compared to other methods. This shows that the small setting of the initial learning rate is correct for almost adaptive methods.

Secondly, we combined Figures 4 and 5a,b to conclude the robustness of each of the methods to network architectures and learning rates. (1) It is observed from Figure 4c that when the learning rate is small, all methods have a similar robustness to network architectures. Specifically, they perform poorer on LeNet and MobileNet than others. The reason is that these two architectures are relatively simple and more sensitive to inputs. However, it is observed from Figure 4a that when the learning rate is large, except Adadelta, Adacomp, and SGD, all other methods are sensitive to network architectures. (2) It is observed from Figure 5a,b that Momentum, Adagrad, RMSprop, Adam, and Adamax are more sensitive to learning rate than others. The reason is that when the learning rate is large, the acceleration in Momentum is too large to diverge, and also the accumulation of gradients in Adagrad, RMSprop, Adam, and Adamax is too large. In contrast, SGD, Adadelta, and Adacomp are relatively insensitive to learning rate, but SGD and Adadelta appear to have an overall degradation when the learning rate is small.

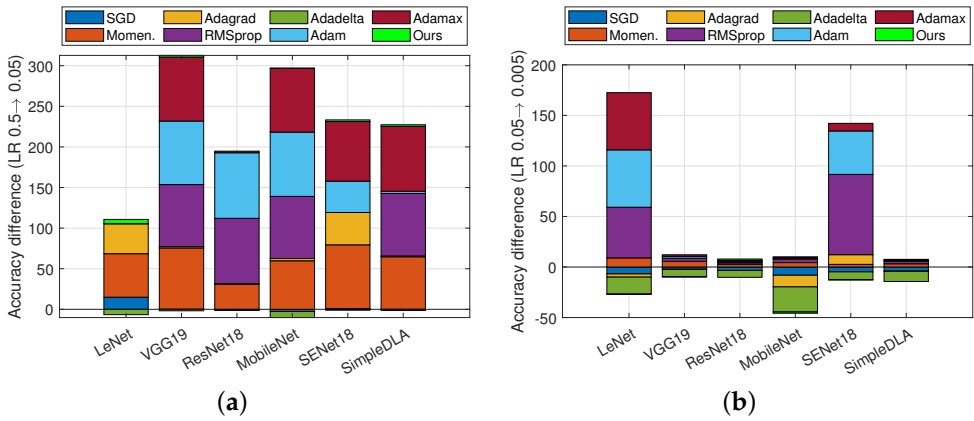

**Figure 5.** Accuracy difference of eight methods when learning rate is changed. Dataset is CIFAR-100. (**a**) Learning rate is changed from 0.5 to 0.05. (**b**) Learning rate is changed from 0.05 to 0.005.

In summary, we conclude that the proposed Adacomp is robust both to learning rates and network architectures.

### 4.4. Results for other Datasets

We conducted more experiments to further validate the robustness of Adacomp to datasets. In particular, we employed the network architecture that was used for MNIST in Section 4.2 but to datasets KMNIST and Fashion-MNIST. Furthermore, we employed a new network architecture (refer to https://github.com/junyuseu/pytorch-cifar-models.git, (accessed on 20 August 2021)) to fulfill the CIFAR-100 classification task. In each case, the learning rate was set as $0.001, 0.01, 0.1, 1$ and we recorded the average time of CIFAR-100 after repeating 10 times. The final prediction accuracy after 150 epochs and the average time on CIFAR-100 are shown in Table 3. Moreover, the average and stand variance of prediction accuracy when the learning rate was $0.001, 0.01, 0.1, 1$ are demonstrated in Figure 6. Three conclusions are observed as follows.

**Table 3.** Prediction accuracy of all methods on three datasets, KMNIST, Fashion-MNIST, and CIFAR-100, is listed, where learning rate was set as $0.001, 0.01, 0.1, 1$. The average time of each method is also recorded, where Tr. and To. are abbreviations for training and total time, respectively.

| Dataset | KMNIST | | | | Fashion-MNIST | | | | CIFAR-100 | | | | |
|---|---|---|---|---|---|---|---|---|---|---|---|---|---|
| lr | 0.001 | 0.01 | 0.1 | 1 | 0.001 | 0.01 | 0.1 | 1 | 0.001 | 0.01 | 0.1 | 1 | Tr./To. Time (s) |
| SGD | 87.43 | 92.74 | **95.07** | 82.36 | 83.31 | 88.60 | 92.26 | 10.00 | 17.91 | 50.82 | 57.69 | 60.59 | 1004.3/1145.5 |
| Momen. | 93.20 | **94.96** | 83.54 | 10.00 | 91.11 | **92.24** | 86.14 | 10.00 | 51.63 | 60.57 | 61.63 | 59.07 | 1052.4/1196.1 |
| Adagrad | 92.02 | 94.34 | 10.00 | 10.00 | 89.36 | 91.92 | 90.58 | 10.00 | 25.47 | 52.77 | 58.23 | 53.04 | 1017.0/1158.0 |
| RMSprop | 93.78 | 10.00 | 10.00 | 10.00 | 91.79 | 10.00 | 10.00 | 10.00 | 59.70 | 58.63 | 29.17 | 1.00 | 1042.0/1185.9 |
| Adadelta | 79.65 | 91.99 | 94.58 | **95.40** | 83.15 | 88.31 | **92.45** | 92.46 | 11.48 | 35.77 | 54.79 | 60.02 | 1025.2/1166.5 |
| Adam | **95.35** | 92.10 | 10.00 | 10.00 | 92.06 | 88.62 | 10.00 | 10.00 | 61.09 | 61.04 | 25.08 | 1.00 | 1030.5/1170.9 |
| Adamax | 95.09 | 94.70 | 10.00 | 10.00 | **92.52** | 91.10 | 86.82 | 10.00 | 57.51 | 61.41 | 58.37 | 1.00 | 1033.6/1176.0 |
| **Ours** | 93.96 | 91.90 | 93.51 | 93.11 | 89.94 | 90.88 | 90.24 | 89.25 | **63.96** | **62.14** | **63.50** | **63.50** | **1010.1/1148.9** |

First, each method has its effective learning rate setting. For example, as Table 3 shows, Adam and Adamax are more effective when the learning rate value is relatively small (e.g., 0.001). However, SGD and Adadelta are more effective when the learning rate value is relatively large (e.g., 0.1). However, when learning rate equals 1, all methods except *Adadelta* and *Adacomp* fail on at least one of the three datasets. Second, as Figure 6 shows, Adacomp is more robust than all other methods on three datasets. In Figure 6, points with large horizontal value mean a high average prediction accuracy and with a small

vertical value mean a low variance. Then, it is obvious to observe that Adacomp is more robust to datasets than other methods. *Third*, Adacomp is computationally efficient. This is directly observed from the last column of Table 3, where the training time and total time are presented. This is because Adacomp only adjusts learning rate according to training loss, while other adaptive methods exploit gradients or additional model parameters.

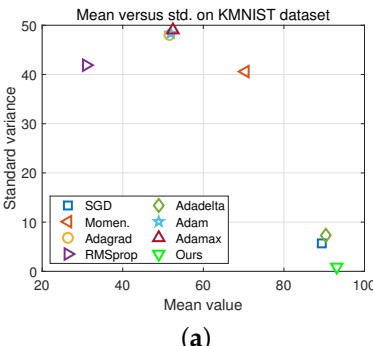 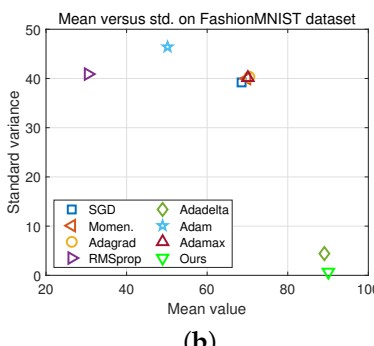 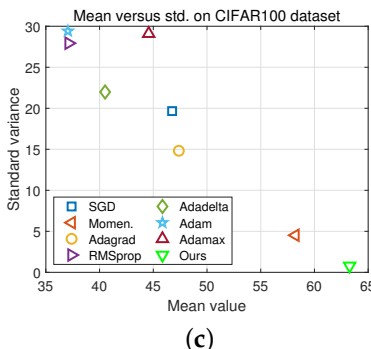

| (**a**) | (**b**) | (**c**) |
|---------|---------|---------|

**Figure 6.** Stability comparison of eight methods on three datasets, KMNIST, Fashion-MNIST, and CIFAR100. Horizontal axis denotes the average prediction value of each method when learning rate was set as 0.001, 0.01, 0.1, 1, and vertical axis denotes the corresponding standard variance. (**a**) Comparison on KMNIST Dataset; (**b**) Comparison on Fashion-MNIST dataset; (**c**) Comparison on CIFAR-100 dataset.

## 5. Conclusions and Future Work

We proposed a method, Adacomp, to adaptively adjust learning rate by only exploiting the values of the loss function. Therefore, Adacomp has higher computational efficiency than other gradient-based adaptive methods, such as Adam and RMSprop. From a high abstract level, Adacomp penalizes large learning rates to ensure the convergence and compensates small learning rates to accelerate the training process. Therefore, Adacomp can help escape from local minima with a certain probability. Extensive experimental results show that Adacomp is robust to network architecture, network initialization, batch size, and learning rate. The experimental results show that Adacomp is inferior to Adadelta and others in the maximum validation accuracy over learning rate. Thus the presented algorithm cannot be an alternative to the state-of-the-art. However, the adaptive methods use different learning rates for different parameters and Adacomp determines only the global learning rate.

In future work, we will validate Adacomp using more tasks (besides convolutional neural networks) and extend Adacomp to a per-dimension first-order algorithm to improve the accuracy of SGD. Additionally, we will apply Adacomp to the distributed environments. Note that Adacomp only uses values of loss functions. Therefore, it is suitable for distributed environments where communication overhead is a bottleneck. Furthermore, we will study how to set the parameters of Adacomp in an end-to-end mode. This may be achieved by introducing feedback and control modules.

**Author Contributions:** Conceptualization, Y.L. and X.R.; methodology, Y.L.; software, F.Z.; validation, Y.L. and S.Y.; formal analysis, Y.L. and F.Z.; writing—original draft preparation, Y.L.; writing—review and editing, X.R. and S.Y. The authors contributed equally to this work. All authors have read and agreed to the published version of the manuscript.

**Funding:** This research received no external funding.

**Institutional Review Board Statement:** Not applicable.

**Informed Consent Statement:** Not applicable.

**Data Availability Statement:** Not applicable.

**Acknowledgments:** This work was supported in part by the National Key Research and Development Program of China under Grant 2020-YFA0713900; in part by the National Natural Science Foundation of China under Grant 61772410, Grant 61802298, Grant U1811461, and Grant 11690011; in part by the China Postdoctoral Science Foundation under Grant 2020T130513 and Grant 2017M623177; and in part by the Fundamental Research Funds for the Central Universities under Grant xjj2018237.

**Conflicts of Interest:** The authors declare no conflict of interest.

## Appendix A. Extended Experiments

More experiments were conducted in the section to validate Adacomp from three aspects.

- Setting different values of $\beta$ in Adacomp to show its impacts. As shown in Equation (9), $\beta$ is a parameter used for tuning the tradeoff between $II_1$ and $II_{21}, II_{22}$. In Section 4, $\beta$ was set as a constant 0.6 for all experiments, without presenting the impacts of $\beta$. Here, experimental results on MNIST show that Adacomp is not so sensitive to $\beta$.
- Using more metrics to provide an overall validation. In Section 4, we used prediction accuracy to compare algorithms for classification tasks. The single metric may not provide the overall validation. Here, we provided complementary experiments with three more metrics, precision, recall, and F1-score. Experimental results on MNIST, Fashion-MNIST, and CIFAR-10 show that Adacomp performs stably under the additional metrics.
- Comparing with evolutionary algorithms to enrich experiments. In Section 4, we compared Adacomp with six first-order adaptive algorithms, lacking comparison with some state-of-the-art approaches. For completeness, we compared Adacomp with two typical evolutionary methods, the genetic algorithm and particle swarm optimization algorithm. Experimental results show that Adacomp can significantly save time cost at the sacrifice of little accuracy degradation.

Experimental details and results are as follows.

### Appendix A.1. Impacts of $\beta$ on Adacomp

As in the discussion about Equation (9), the range of $\beta$ is $(0.5, 5]$. To present a comprehensive investigation, we selected $\beta$ from $[0.6, 1]$ with step size 0.1 and from $[1, 5]$ with step size 0.5. Furthermore, learning rate was set as $0.01, 0.1, 1$, and other settings were the same as in Figure 2. Detailed results are shown in Figure A1, from which two conclusions are obtained. For simplicity, *lr* is an abbreviation for learning rate and $c$ in the parentheses denotes clipping gradient with bound 10.

*First*, Adacomp is more sensitive to $\beta$ when the learning rate is small. When $lr = 1$, Adacomp fails at two settings $\beta = 4, 5$, and the number of failing settings decreases with *lr*. When $lr = 0.01$, the effective $\beta$ almost lies in the range $[0.6, 1]$. This shows that Adacomp is insensitive to learning rate when $\beta \leq 1$. *Second*, clipping gradient can significantly improve the robustness of Adacomp. Based on the training process, we found that the divergence happened at the early few iterations where the gradient norm tended to infinity. To overcome this, we clipped gradients with bound 10 (larger or less has no impact). Results when $lr = 0.01(c), 0.1(c), 1(c)$ show that Adacomp is robust to both $\beta$ and learning rate when additionally clipping the gradient.

### Appendix A.2. Comparison Using More Metrics

To present an overall validation, we adopted three additional metrics, precision, recall (also sensitivity), and F1-score (weighted average of precision and recall) in the section. The datasets used include MNIST, Fashion-MNIST, and CIFAR-10. The learning rate was set as $0.01, 0.1, 1$. For MNIST and Fashion-MNIST, the network architecture and parameter settings were same as the corresponding experiments in Section 4. For CIFAR-10, the network architecture was LeNet. Detailed results are shown in Table A2, where precision, recall, and F1-score are average on all classes. Two conclusions are observed.

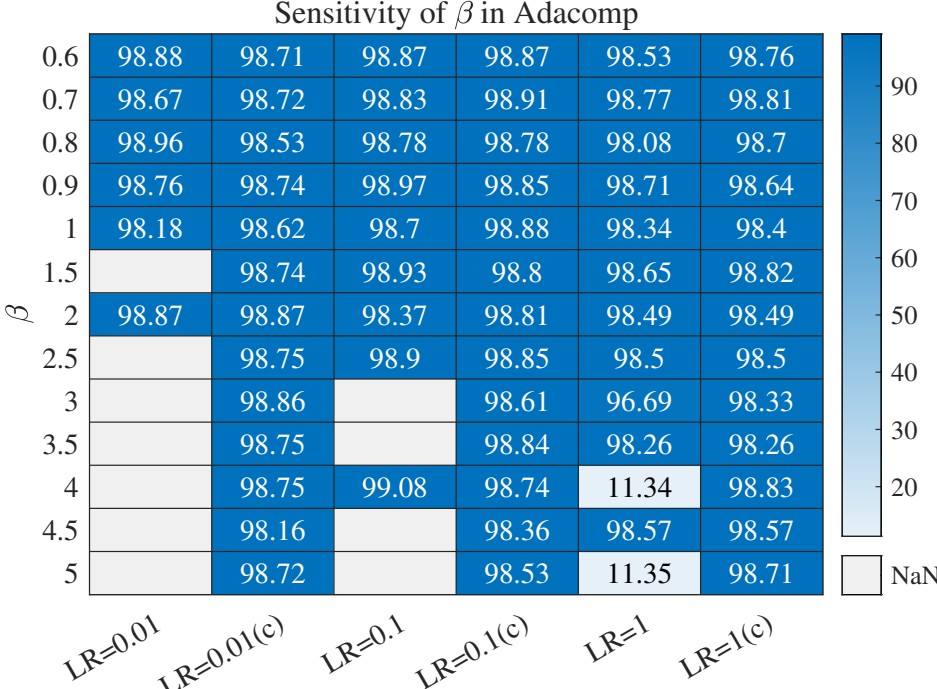

**Figure A1.** Impacts of $\beta$ of Adacomp on prediction accuracy. The dataset used was MNIST and the network architecture was as the same as in Section 4.2. Here LR is an abbreviation for learning rate and *c* in the parentheses denotes gradients are clipped.

First, each algorithm has a similar precision, recall, and F1-score when accuracy is high. For example, when accuracy is greater than 90% on MNIST (or 80% on Fashion-MNIST, 60% on CIFAR-10), the performance under different metrics almost remains unchanged. This is mainly because these datasets have even class distribution. Second, although Adacomp does not always achieve the highest accuracy or F1-score, it is the most robust to learning rate. For a large learning rate ($lr = 1$), only Adadelta and Adacomp (ours) work well. Within the two algorithms, Adacomp is more robust than Adadelta. Note that on CIFAR-10, Adadelta achieves 0.45, 0.61, 0.65 F1-score when $lr = 0.01, 0.1, 1$, respectively. Meanwhile, Adacomp achieves 0.65, 0.67, 0.65 F1-score. This can be also observed on Fashion-MNIST.

*Appendix A.3. Comparison with Evolutionary Algorithms*

We compared Adacomp with two evolutionary algorithms, the genetic algorithm (GA) and particle swarm optimization algorithm (PSO), on MNIST, Fashion-MNIST, and CIFAR-10 datasets. The code for GA was modified based on https://github.com/jishnup11/-Fast-CNN-Fast-Optimisation-of-CNN-Architecture-Using-Genetic-Algorithm, (accessed on 24 October 2021 ) and the code for PSA was based on https://github.com/vinthony/pso-cnn, (accessed on 24 October 2021). Specific parameter settings were as follows and detailed results are shown in Table A1, where results for Adacomp are the highest values corresponding to Table A2.

- For the GA, we set 10 generations with 20 populations in each generation. Mutation, random selection, and retain probability were set as 0.2, 0.1, 0.4, respectively.
- For PSO, swarm size and number of iterations were both set as 100, inertia weight and acceleration coefficients were both set as 0.5.

Two conclusions are obtained from Table A1. *First*, GA and PSO outperform Adacomp on MNIST and Fashion-MNIST. This is because GA and PSO explore too many parameter settings compared to Adacomp. However, this does not hold for CIFAR-10. Therefore, GA and PSO have a high probability of finding the better solutions by exploring a larger space.

*Second,* Adacomp is more efficient than GA and PSO. For example, Adacomp performed similarly to GA and PSO; however, the consumed time was only about 1/10 compared to them.

Based on Table A2, it may be possible to combine GA or PSO with Adacomp to simultaneously improve the efficiency and utility.

**Table A1.** Comparison results of Adacomp and two evolutionary algorithms are presented, where GA and PSO are abbreviations for the genetic algorithm and particle swarm optimization algorithm, respectively.

| | Dataset | MNIST | | | | Fashion-MNIST | | | | CIFAR-10 | | | |
|---|---|---|---|---|---|---|---|---|---|---|---|---|---|
| | Metrics | Acc. | Prec. | Reca. | F1-sc. | Acc. | Prec. | Reca. | F1-sc. | Acc. | Prec. | Reca. | F1-sc. |
| GA [52] | Maximum | 99.16 | 99.19 | 99.16 | 99.17 | 92.22 | 92.56 | 91.96 | 92.25 | 61.45 | 62.45 | 61.34 | 61.89 |
| | Time | | 32 m/110 m | | | | 58 m/71 m | | | | 5.8 h/5.8 h | | |
| PSO [53] | Maximum | 99.11 | 99.16 | 99.09 | 99.12 | 91.99 | 92.48 | 91.72 | 92.09 | 65.41 | 65.96 | 65.39 | 65.67 |
| | Time | | 23 m/66 m | | | | 16 m/34 m | | | | 3.0 h/6.5 h | | |
| Ours | Maximum | 98.45 | 98.45 | 98.44 | 98.44 | 89.10 | 89.05 | 89.10 | 89.05 | 67.45 | 67.09 | 67.45 | 67.26 |
| | Time | | 2.57 m–2.6 m | | | | 2.7 m–2.75 m | | | | 0.63 h–0.64 h | | |

**Table A2.** Comparison results of Adacomp between other adaptive methods with more evaluation metrics are presented, where learning rate was set as LR = 0.01, 0.1, 1. Acc., Pre., Reca., and F1-sc. are abbreviations for accuracy, precision, recall, and F1-score, respectively.

| | Dataset | MNIST | | | | Fashion-MNIST | | | | CIFAR-10 | | | |
|---|---|---|---|---|---|---|---|---|---|---|---|---|---|
| LR | Metrics | Acc. | Pre. | Reca. | F1-sc. | Acc. | Pre. | Reca. | F1-sc. | Acc. | Pre. | Reca. | F1-sc. |
| 0.01 | SGD | 97.71 | 97.13 | 97.10 | 97.11 | 88.30 | 88.24 | 88.30 | 88.25 | 60.11 | 59.72 | 60.12 | 59.72 |
| | Moment. | 98.78 | 98.77 | 98.78 | 98.77 | 91.60 | 91.61 | 91.60 | 91.62 | 67.16 | 67.00 | 67.16 | 67.01 |
| | Adagrad | 98.90 | 98.90 | 98.89 | 98.89 | 91.95 | 91.93 | 91.95 | 91.93 | 62.71 | 62.54 | 62.71 | 62.44 |
| | RMSprop | 11.24 | 2.16 | 10.00 | 3.55 | 87.78 | 87.80 | 87.78 | 87.76 | 47.91 | 47.52 | 47.91 | 47.57 |
| | Adadelta | 96.40 | 96.38 | 96.36 | 96.37 | 86.55 | 86.44 | 86.55 | 86.47 | 47.00 | 46.34 | 47.00 | 45.89 |
| | Adam | 97.85 | 97.85 | 97.83 | 97.84 | 88.00 | 88.03 | 88.00 | 88.01 | 10.00 | 9.99 | 10.00 | 9.93 |
| | Adamax | 98.64 | 98.63 | 98.63 | 98.63 | 90.11 | 90.10 | 90.11 | 90.10 | 66.38 | 66.19 | 66.38 | 66.23 |
| | Ours | 98.08 | 98.08 | 98.07 | 98.07 | 88.89 | 88.85 | 88.89 | 88.85 | 65.36 | 64.88 | 65.36 | 65.11 |
| 0.1 | SGD | 98.86 | 98.86 | 98.85 | 98.86 | 91.33 | 91.29 | 91.33 | 91.30 | 65.82 | 65.60 | 65.81 | 65.64 |
| | Moment. | 97.83 | 97.82 | 97.81 | 97.81 | 87.88 | 88.03 | 87.88 | 87.93 | 44.71 | 44.12 | 44.71 | 44.25 |
| | Adagrad | 96.97 | 96.95 | 96.95 | 96.95 | 88.68 | 88.62 | 88.68 | 88.63 | 53.02 | 52.58 | 53.02 | 52.62 |
| | RMSprop | 10.78 | 5.11 | 9.99 | 6.76 | 11.36 | 17.58 | 11.36 | 13.80 | 10.00 | 10.00 | 10.00 | 9.62 |
| | Adadelta | 98.80 | 98.79 | 98.79 | 98.79 | 91.15 | 91.12 | 91.15 | 91.12 | 62.07 | 61.89 | 62.07 | 61.85 |
| | Adam | 10.81 | 21.63 | 10.00 | 13.67 | 10.00 | 7.99 | 10.00 | 8.88 | 10.00 | 10.00 | 10.00 | 9.63 |
| | Adamax | 10.65 | 4.13 | 9.99 | 5.84 | 84.36 | 84.35 | 84.36 | 84.34 | 10.00 | 9.99 | 10.00 | 9.64 |
| | Ours | 97.92 | 97.91 | 97.91 | 97.91 | 88.90 | 88.86 | 88.90 | 88.86 | 67.45 | 67.09 | 67.45 | 67.26 |
| 1 | SGD | 9.80 | 0.98 | 10.00 | 1.78 | 10.00 | 1.00 | 10.00 | 1.81 | 10.00 | 10.00 | 10.00 | 10.00 |
| | Moment. | 9.80 | 0.98 | 10.00 | 1.78 | 10.00 | 10.00 | 10.00 | 10.00 | 10.00 | 9.99 | 9.99 | 9.77 |
| | Adagrad | 10.97 | 3.13 | 9.99 | 4.76 | 62.34 | 70.20 | 62.34 | 64.65 | 10.00 | 10.00 | 10.00 | 9.34 |
| | RMSprop | 10.01 | 6.06 | 9.99 | 7.54 | 10.10 | 10.00 | 10.10 | 9.63 | 10.00 | 9.99 | 9.99 | 9.71 |
| | Adadelta | 99.02 | 99.01 | 99.01 | 99.01 | 91.91 | 91.91 | 91.91 | 91.91 | 65.26 | 65.22 | 65.26 | 65.19 |
| | Adam | 10.15 | 40.75 | 10.00 | 16.05 | 10.00 | 9.99 | 10.00 | 9.73 | 10.00 | 9.99 | 9.99 | 9.74 |
| | Adamax | 10.57 | 4.15 | 10.00 | 5.86 | 9.99 | 8.99 | 9.99 | 9.46 | 10.00 | 9.99 | 10.00 | 9.78 |
| | Ours | 98.45 | 98.45 | 98.44 | 98.44 | 89.10 | 89.05 | 89.10 | 89.05 | 65.67 | 65.56 | 65.67 | 65.61 |

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
