# Peer review of "A Zeroth-Order Adaptive Learning Rate Method to Reduce Cost of Hyperparameter Tuning for Deep Learning"

_applsci, doi:10.3390/app112110184_

Round 1
Reviewer 1 Report
This paper proposed a method to adjust learning rate using values of loss function. This feature can be effective in terms of computational runtime efficiency. The proposed method penalizes large learning rate to ensure the convergence and compensates small learning rate to accelerate the training process. According to the tuning results, the performance of the method can be considerable. However, there are some major deficiencies as follows.
- In the abstract, the main challenges should be described briefly and then the main proposed solutions should be mentioned. It is not necessary to discuss the technical details of the research gaps.
- In the introduction, please rewrite the main contributions of this paper and the research question, and motivations. what are the main advantages to select these two models to solve this problem? Furthermore, in this section, initially please list the research questions and hypothesises.
- In the section of Proposed method, please discuss how did you select the range of hyperparameters?
- The literature review is poor and is not summerized comprehensively. please consider the newly published works in the field of deep learning hyper-parameters such as A deep learning-based evolutionary model for short-term wind speed forecasting: A case study of the Lillgrund offshore wind farm. Energy Conversion and Management, 236, 114002.Hyperparameter tuning deep learning for diabetic retinopathy fundus image classification. IEEE Access, 8, 118164-118173. Combination of hyperband and Bayesian optimization for hyperparameter optimization in deep learning. arXiv preprint arXiv:1801.01596.
- Please describe the Algorithm 2 with more technical details.
- For Adacomp Method section, there are several vague sentences, please focus on the proposed equations and describe them with more details such as all variables should be listed in the text and represent.
Author Response
Thanks a lot for the useful comments.
Please see the attachment for detailed revisions.

Reviewer 2 Report
This paper propose a heuristic zeroth-order learning rate method, Adacomp, to reduce the cost of tuning hyperparameters.
I would recommend to add other evaluation metrics, currently only accuracy has been added. It is important to show performance on other evaluation metrics
What is the latency to train the model?
There is lack of comparision with state-of-the-art approaches
I would recommend to compare the current model with genetic algorithm and particle swarm optimization
Author Response
Thanks a lot for the meaningful comments.
Please see the attachment for detailed revisions.

Reviewer 3 Report
Summary
The paper proposes an adaptive learning-rate (LR) method, called Adacomp, that determines the global LR for stochastic gradient descent (SGD) using the loss in a zeroth-order manner. Since it reflects the objective loss function, the algorithm is robust to the hyperparameters including the initial LR, the batch size, and the network initialization. The presented algorithm is analyzed and explained in detail. Experimental results using the major benchmarks are also presented. Based on the experimental results, I think Adacomp cannot be an alternative to the state-of-the-art optimization methods at this time. Nevertheless, the presented algorithm would be of interest to readers of the Applied Sciences Journal.
Strong points
1. Adacomp is shown to be quite robust to the choice of learning-rate. (I suppose the LR using Adacomp method rises from the initial LR to a high value in the first few epochs before shrinking to small value. This is why Adacomp is robust to the initial LR. If so, the authors can mention this characteristic in paper.)
2. The methodology of the presented algorithm is carefully derived and analyzed. The two ideas presented in this paper and its implementation have sufficient novelty.
3. The paper is well written and structured; the related studies are summarized in detail; and the code is supplied for reproducibility and accessibility.
Major concerns
1. It is not clear how much Adacomp depends on the parameter $\beta$. I would strongly recommend presenting the accuracy over varying $\beta$ under a fixed condition (MNIST, ConvNet, and LR=0.01 for example). Even if Adacomp is sensitive to $\beta$, the presented algorithm has a practical value since it reduces the number of hyper-parameters from three (LR, batch-size, and initialization) to one.
2. The experimental results show that Adacomp is inferior to the Adadelta and others in the maximum validation accuracy over LR. Thus the presented algorithm cannot be an alternative to the state-of-the-art. However, the adaptive methods use different LRs for different parameters and Adacomp determines only the global LR. I think the authors can emphasize this difference in discussion. Moreover, I suppose the idea of Adacomp can be extended in future work to a per-dimension 1st-order algorithm that will improve the accuracy of SGD.
3. In line 232, the authors reveal that Adacomp can fluctuate at a local minima. Please explain more about this limitation. Is it derived from the equations? Or observed experimentally?
Minor concerns
Lines 24-25: Please show the definition of the term f*.
Line 240: two classification datasets. Typo?
Line 251: the used network architecture. Is it the ConvNet? Please describe more concretely.
Lines 294-295: What is the initialized seed? If it is the random seed, the specific values 1, 10, 30, and 50 are not informative. The authors can say “Figure 3(b) shows the impact when we have repeated the experiment four individual times in which the network is initialized using the same random seed for different optimizers.” for example.
Line 332: Typo.
Line 364: The title of Section 4.4 can be revised as ‘Results on other Datasets’.
Lines 338, 396: How about removing the paragraph titles.
Best regards,
Author Response

(The authors gave the same response as above.)

Round 2
Reviewer 1 Report
The applied modifications are appropriate to publish this paper.